# Effects of Alfalfa Hay to Oat Hay Ratios on Chemical Composition, Fermentation Characteristics, and Fungal Communities during Aerobic Exposure of Fermented Total Mixed Ration

Mingjian Liu [1,2,3], Lin Sun [4], Zhijun Wang [1,2,3], Gentu Ge [1,2,3], Yushan Jia [1,2,3,*] and Shuai Du [1,2,3,*]

1   College of Grassland, Resources and Environment, Inner Mongolia Agricultural University,
    Hohhot 010019, China; liumj_nm@163.com (M.L.); zhijunwang321@126.com (Z.W.); gegentu@126.com (G.G.)
2   Key Laboratory of Forage Cultivation, Processing and High Efficient Utilization, Ministry of Agriculture,
    Hohhot 010019, China
3   Key Laboratory of Grassland Resources, Ministry of Education, Hohhot 010019, China
4   Inner Mongolia Academy of Agricultural and Animal Husbandry Sciences, Hohhot 010031, China;
    sunlin2013@126.com
*   Correspondence: jys_nm@sina.com (Y.J.); nmgdushuai@zju.edu.cn (S.D.)

**Abstract:** The application of fermented total mixed ration (FTMR) is an effective method to prolong the use time of feed, but the understanding of the interaction mechanism between fungal microorganisms and silage quality and aerobic stability in FTMR is still limited. This study aimed to evaluate the effects of alfalfa (*Medicago sativa* L.) hay to oat (*Avena sativa* L.) hay ratios on chemical composition, fermentation characteristics, and fungal communities during aerobic exposure of fermented total mixed ration (FTMR). The supplement levels of oat were as follows: 200 g/kg oat hay (LO), 300 g/kg oat hay (MO), and 400 g/kg oat hay (HO). The water content of the three treatments was adjusted to 50% using a sprayer. After 60 days of ensiling, the bags were opened, and the chemical composition, fermentation characteristics, and fungal communities were measured after 3, 6, 9, and 12 days of aerobic exposure. The results suggested that the LO treatment significantly ($p < 0.05$) increased the aerobic stability than that in other treatments. The crude protein and lactic acid content in the three treatments were significantly decreased with the extension of the aerobic exposure period. Additionally, there was a remarkable ($p < 0.05$) higher lactic acid content observed in the LO treatment than that in the HO treatment during the aerobic stage. The PCoA showed that the compositions of the fungal community in the HO treatment were distinctly separated from the other two treatments. Compared with HO and LO treatments, the MO treatment observed relatively higher OTU, Shannon, and Chao1 indexes. Compared with LO and MO treatments, the abundances of the genes *Saccharomyces* and *Wallemia* were greater increased and decreased in the HO treatment, respectively. Integrated correlation analysis also underscores a possible link between the fermentation characteristics, aerobic stability, and significantly altered fungal community. This study suggested that the use of FTMR in production might prolong aerobic storage time when alfalfa was fermented in a mixture with ≤30% oat.

**Keywords:** oat; alfalfa; fermentation characteristics; fungal communities; aerobic stability; fermented total mixed ration

## 1. Introduction

Alfalfa (*Medicago sativa* L.) hay is generally considered the main feed source for domestic ruminants, which is vital and guaranteed to supply high-quality ruminant products due to its high protein concentration and digestible fiber [1]. However, the sustainable availability of alfalfa hay is a challenging task because of the high cost of alfalfa hay production [2]. Therefore, suitable alternative forages need to be urgently identified to meet the needs of the herbivorous animal industry. Oat (*Avena sativa* L.) is widely grown in the world, and the lower

production cost and higher digestibility make it ideal to become a high-quality feed crop to replace alfalfa hay for animal production [3]. Meanwhile, increasing the oat content of the diet will provide sufficient physically active fiber to increase the relative content and activity of rumen anaerobic fungi, which in turn will promote fiber degradation [4–6]. Additionally, a previous study has shown that some fungi in silage, such as *Pichia*, could survive in the rumen environment and enhance the rumen degradation of fiber [7]. Sun et al. [8] also reported that the mixture of oat hay and alfalfa hay in the TMR (total mixed ration) feeding system could greatly promote the growth performance of lambs.

The TMR feeding system was widely used in animal production around the world due to the advantages of precision, efficiency, and convenience [9]. A previous study clarified that the TMR could balance nutrition and rumen stability for the animals relative to the conventional feeding systems in ruminants [10]. However, the proliferation of harmful microorganisms easily occurs due to the wet and crumbled particularity of TMR, resulting in aerobic deterioration of TMR [11,12]. To overcome this disadvantage, the practice of combining TMR and silage technology to prepare fermented total mixed ration (FTMR) has become an effective method to prolong the storage time and improve the aerobic stability of TMR in many countries [13]. Hao et al. [14] reported that the FTMR has considerable resistance to aerobic deterioration compared with TMR. In general, there are many factors affecting aerobic stability, such as undissociated acetic acid in plants and the development of yeasts and molds. However, recent research has demonstrated that the improvement of aerobic stability in silage is closely related to dynamic changes in microorganisms during the aerobic exposure stage [15]. The composition and distribution of fugal are the key players in aerobic stability. During the aerobic exposure stage, the growth of undesirable microorganisms could lead to the deterioration of nutrients in silage and modify the aerobic stability [16]. Weinberg et al. [17] showed that the presence of fungi in raw materials and silages could directly influence the accumulation of mycotoxins and aerobic stability. Liu et al. [16] reported that the aerobic stability of *Leymus chinensis* silage was affected by a variety of fungi. Therefore, the effect of the fungal community on the quality of FTMR during aerobic exposure cannot be ignored, and it is crucial to fully understand the interaction mechanisms between fungal microorganisms and the quality of silage for regulating aerobic stability and improving the utilization of silage. However, to the best of our knowledge, there is very little information that reveals the interaction mechanism between fungal microorganisms and the silage quality in FTMR during the aerobic exposure stage.

The purpose of this study was to evaluate the effect of replacing alfalfa hay with oats in FTMR on silage quality and fungal communities during the aerobic exposure stage and reveal the interaction mechanism between fungal microorganisms and silage quality and aerobic stability in FTMR.

## 2. Materials and Methods

### 2.1. FTMR Preparation

All raw materials in this experiment were provided by Chaoyue Feed Co., Ltd. (Balin Left Banner, Chifeng, China). The supplement levels of oat (contains no grains) were as follows: 200 g/kg oat hay (LO), 300 g/kg oat hay (MO), and 400 g/kg oat hay (HO). The FTMR was prepared as follows: The roughage was chopped into approximately 1–2 cm and mixed with concentrate feed according to the above ratio. A compound bacterial agent (the bacteria compound contains bacillus subtilis R2 and bacillus subtilis N10) was inoculated in the FTMR (1 g/kg of fresh TMR), and which was purchased from Hebei Zhong bang Biotechnology Co., Ltd. (Strong brand, Cangzhou, China). The additive was dissolved in water, and the water content of the three treatments was adjusted to 50% by using a sprayer. After the application of the treatments, the forages were mixed thoroughly with additives, placed within fermentation bags (55 cm × 85 cm) with a one-way exhaust valve, compacted, and sealed at the bag mouth. There were five repetitions of each group, and all samples were stored indoors at 15 °C for approximately 60 days for fermentation. The ingredient compositions of the FTMR in three treatments are illustrated in Table 1.

**Table 1.** Ingredients and chemical composition of dietary.

| Items | LO | MO | HO | SEM | *p*-Value |
|---|---|---|---|---|---|
| Ingredient (g/kg DM) | | | | | |
| Oat hay | 200 | 300 | 400 | - | - |
| Alfalfa hay | 400 | 300 | 200 | - | - |
| Natural forage | 30 | 30 | 30 | - | - |
| Corn stalk | 20 | 20 | 20 | - | - |
| Corn | 220 | 200 | 180 | - | - |
| Soybean meal | 90 | 110 | 130 | - | - |
| Wheat bran | 20 | 20 | 20 | - | - |
| Calcium hydrogen phosphate | 3 | 3 | 3 | - | - |
| NaCl | 2 | 2 | 2 | - | - |
| NaHCO$_3$ | 5 | 5 | 5 | - | - |
| Premix | 10 | 10 | 10 | - | - |
| Chemical compositions | | | | | |
| DM (g/kg FW) | 63.17 | 63.34 | 63.21 | 0.2528 | 0.9241 |
| CP (g/kg DM) | 13.33 | 13.28 | 13.12 | 0.1107 | 0.7873 |
| NDF (g/kg DM) | 43.20 c | 50.29 b | 55.30 a | 1.7237 | 0.0005 |
| ADF (g/kg DM) | 29.53 | 31.83 | 33.77 | 0.8122 | 0.1191 |
| Fermentation profile | | | | | |
| pH | 4.55 | 4.52 | 4.49 | 0.0243 | 0.7547 |
| Lactic acid (g/kg DM) | 10.28 | 9.86 | 9.69 | 0.2635 | 0.7329 |
| Acetic acid (g/kg DM) | 1.15 a | 0.88 b | 0.76 b | 0.0651 | 0.0279 |
| Propionic acid (g/kg DM) | 1.24 a | 0.63 c | 0.94 b | 0.0914 | 0.0055 |
| Ammonia-N (g/kg DM) | 3.03 b | 2.65 b | 4.39 a | 0.2556 | 0.0001 |
| Microbial counts | | | | | |
| Lactic acid bacteria (Log$_{10}$ cfu/g FM) | 5.90 b | 7.81 a | 5.91 b | 0.3183 | 0.0015 |
| Aerobic bacteria (Log$_{10}$ cfu/g FM) | 6.05 a | 4.48 b | 6.74 a | 0.3433 | 0.0009 |

DM, dry matter; CP, crude protein; NDF, neutral detergent fiber; ADF, acid detergent fiber. SEM, standard error of means. Composition of mineral premix. Per kg: Copper 1800 mg, iron 3400 mg, manganese 1500 mg, zinc 1700 mg, cobalt 20 mg, vitamin A 1620 000 IU, vitamin D332 400 IU, vitamin E 540 IU, folic acid 15 mg. LO, low oat percentages group; MO, middle oat percentages group; HO, high oat percentages group. Different lowercase letters indicate significant differences among different treatments (*p* < 0.05).

## 2.2. Aerobic Stability

According to the method of [18], the aerobic stability of FTMR was measured after opening the bags. More specifically, the 500 g samples from each treatment were transferred to 1 L sterile bottles (diameter, 12.6 cm; height, 13.0 cm). Aerobic stability was measured by a thermometer inserted in the center of the bottle and recorded at 12 h intervals as described by Amaral et al. [19]. Notably, two layers of cheesecloth covered each container to avoid contamination by impurities and water loss and allowed the air to penetrate. The time taken to exceed the temperature of the FTMR by 2 °C above ambient temperature was defined as aerobic stability [20]. Additionally, the aerobic FTMR samples of each treatment were thoroughly mixed and sampled after 3, 6, 9, and 12 days of aerobic exposure to analyze the chemical composition, fermentation quality, and fungal communities.

## 2.3. Chemical, Fermentation, and Microbial Analyses

FTMR samples (100 g) were taken at 0, 3, 6, 9, and 12 days, and the dry matter (DM) content was calculated by drying the samples in a forced-draft oven at 65 °C for 72 h, then ground in a sample mill equipped with a 1 mm sieve for further analysis of chemical composition. Notably, the DM content was corrected to include the volatile solids according to the formula provided by Haigh [21]. The neutral detergent fiber (NDF) and acid detergent fiber (ADF) contents were calculated using an ANKOM fiber analyzer (Model: A2000i; Beijing Anke Borui Technology Co., Ltd., Beijing, China) following the method reported by Van Soest et al. [22]. The crude protein (CP) content was determined following the AOAC (1990) standard procedures [23].

After 3, 6, 9, and 12 days of aerobic exposure, the samples (10 g) of FTMR were mixed with 90 mL of sterile water and homogenized in a blender for 1 min to extract the fermentation broth, followed by filtration using four layers of gauze, and the filtrate was aliquoted into three centrifuge tubes. Thereafter, a glass electrode pH meter (STARTER 100/B, OHAUS, Shanghai, China) was calibrated and performed to measure the pH value of the first filtrate. An HPLC (1200, Agilent, Santa Clara, CA, USA) with a 210 nm UV detector and a column (ICSep COREGEL-87H) quantified the levels of lactic acid (LA), acetic acid (AA), propionic acid (PA), and butyric acid (BA). The mobile phase was 3 mmol/L $HClO_4$ at a flow rate of 1.0 mL $min^{-1}$ at 50 °C [24]. Similar to the method of Broderick et al. [25], the ammonia nitrogen ($NH_3$-N) content was detected based on colorimetry by using the phenol-hypochlorite method.

The third filtrate was serially diluted ($10^{-1}$ through $10^{-5}$) with sterilized water, and the microbial populations were evaluated according to a previous report [26]. Briefly, the lactic acid bacteria (LAB) numbers were detected with de_Man Rogosa_Sharpe agar (MRS) incubated at 30 °C for 48 h under anaerobic conditions, whereas the aerobic bacteria (AB) and yeasts were grown on nutrient agar and malt extract agar incubated for 24 h at 30 °C under aerobic conditions, respectively. All culture media used in this work were purchased from the same manufacturer (Guangzhou Huankai Microbial Science and Technology Co., Ltd., Guangzhou, China).

## 2.4. DNA Extraction, Amplification, and Sequencing

The total DNA was isolated from fungi in all FTMR samples with an E.Z.N.A. RStool DNA Kit (D4015-04, Omega, Inc., United States) based on the instructions of the manufacturer. The primers of ITS1 specific forward (5′-GTGARTCATCGAATCTTTG-3′) and ITS2 reverse (5′-TCCTCCGCTTATTGATATGC-3′) were used to amplify the V3–V4 region of 18S rRNA gene. The amplified products were purified and recovered using the 1.0% agarose gel electrophoresis method. After purification, the purified PCR amplicons were paired-end sequenced using the Illumina MiSeq PE300 platform (Illumina Inc., San Diego, CA, USA). The sequencing library was prepared using the gDNA samples using the Illumina library quantification kit (Kapa Biosciences, Woburn, MA, USA). The amplicon libraries were evaluated and characterized for size distribution and the number of amplicon libraries using an Agilent 2100 Bioanalyzer (Agilent, Santa Clara, CA, USA). Sequencing data for 18S rRNA gene sequence have been deposited in NCBI, and sequence read archive databases are publicly available at BioProject (PRJNA947788).

## 2.5. Bioinformatics Analysis

The raw sequences were cleaned, merged, and clustered as described by Wang et al. [27]. More specifically, the paired-end reads were assigned to different samples after sequencing due to their unique barcodes and truncated by removing the barcode and primer sequences. Paired-end reads were merged with the FLASH software (v1.2.8). The fqtrim (v0.94) was selected to filter the quality of the original reads to obtain high-quality clean reads, and the Vsearch (v2.3.4) software was performed for filtering the chimeric sequences. Raw reads were processed in QIIME2, and the imported paired reads were quality filtered, denoised, and merged using the plugin DADA2 (v3.11) to generate the ASV feature table. Taxonomic assignment of the ASVs was completed using the ITS database (unite8.0/its_fungi). The taxonomy assignment was double-checked with BLAST suite tools. The alpha (Shannon, Simpson, and Chao1 indices) analysis was analyzed and visualized by the plugin q2-diversity script in QIIME2 (v2019.7). Goods coverage is calculated as coverage = $1 - (S/N)$, where S is the number of unique circular consensus sequences (CCS), and N is the number of individuals in the sample. To compare beta diversity, weighted UniFrac outputs were calculated and visualized using PCoA by R. The stamp analyses were selected in this research to determine taxa that were greatly different among the three treatments. The main differentially abundant genera among the three treatments were visualized during different stages by the linear discrimination analysis coupled with the effect size (LEfSe)

method. The genes' metabolic pathways were predicted with the Fungi Functional Guild (FUNGuild) tool (http://www.funguild.org/ (accessed on 1 January 2023). A mental test correlation heat map between dominant genera and fermentation parameters was generated by R based on Pearson's correlations [28]. The Kyoto Encyclopedia of Genes and Genomes (KEGG) Module database were also selected to assign the genes to enzyme functions. Bar graphs and line charts were drawn with the GraphPad Prism 9 (San Diego, CA, USA).

### 2.6. Statistical Analysis

Analyses of alpha diversity, chemical composition, and fermentation characteristics were performed using SAS ver. 9.2 (SAS Institute, 2007 Cary, NC, USA) [29]. The statistical model of the SAS was as follows: $Y_{ij} = \mu + T_i + D_j + (T \times D)_{ij} + \varepsilon_{ij}$, where $Y_{ij}$ = response variable, $\mu$ = overall mean, $T_i$ = the effect of oat supplement (LO, MO, HO), $D_j$ = aerobic exposure time (3 d, 6 d, 9 d, 12 d), $(T \times D)_{ij}$ = the effect of the interaction between the oat supplement and aerobic exposure time, and $\varepsilon_{ij}$ = random residual error. Differences between means were assessed using the analysis of variance (ANOVA) TukeyHSD post hoc test, and $p < 0.05$ was considered statistically significant.

## 3. Results

### 3.1. Effects of Alfalfa Hay to Oat Hay Ratios on Aerobic Stability during Aerobic Exposure of FTMR

As shown in Figure 1, the temperatures in the three groups were 2 °C above room temperature after 228, 204, and 132 h of aerobic exposure, respectively. The aerobic stability was significantly ($p < 0.05$) lower in the HO group than that in the other two groups.

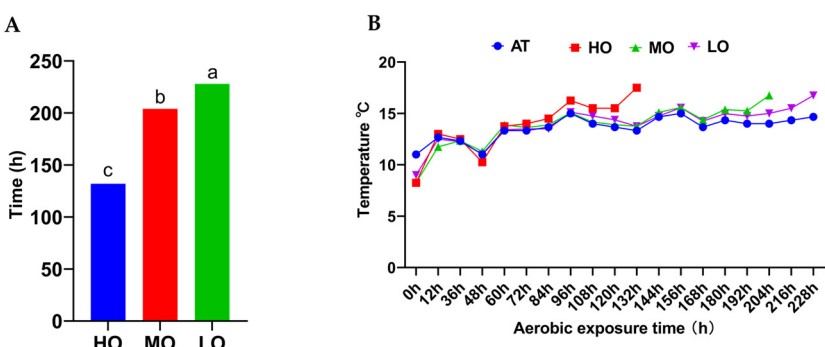

**Figure 1.** Hours of aerobic stability (**A**) and Dynamic changes in temperatures (**B**) of FTMR during air exposure. AT, ambient temperature; LO, low oat percentages treatment; MO, middle oat percentages treatment; HO, high oat percentages treatment. Different lowercase letters indicate significant differences among different treatments ($p < 0.05$).

### 3.2. Fermentation Characteristics of FTMR after Aerobic Exposure

The fermentation characteristics of FTMR after aerobic exposure are shown in Table 2. The pH, NH$_3$-N, and BA content were significantly increased with the extension of the aerobic exposure period. The interaction of oat addition and aerobic exposure time had a significant effect on NH$_3$-N content, and the NH$_3$-N concentration was higher in the HO treatment relative to the LO and MO treatments after 6 and 9 d of aerobic exposure. The interaction of oat addition and aerobic exposure time greatly affected the BA content, which was higher in the LO treatment than that in the MO treatment after 6 d of aerobic exposure ($p < 0.05$), while the highest BA content was observed in the HO treatment compared with LO and MO treatments. Interestingly, all three treatments decreased the contents of LA, AA, and PA with the extension of the aerobic exposure period. Compared with the HO treatment, the LO treatment had a higher LA concentration after 3, 9, and 12 d of aerobic exposure ($p < 0.05$). After 3 d of aerobic exposure, lower AA concentration was observed in the HO treatment than that in the LO treatment after 6, 9, and 12 d of aerobic exposure.

During the whole aerobic exposure period, lower PA concentrations were observed in the MO treatment compared with the LO treatment.

**Table 2.** Fermentation characteristics of FTMR after aerobic exposure.

| Item | Treatment | Aerobic Exposure Days | | | | SEM | *p*-Value | | |
|------|-----------|------|------|------|------|-----|-----|-----|-----|
| | | AE3 | AE6 | AE9 | AE12 | | T | D | D × T |
| pH | LO | 4.79 Ac | 4.87 Ac | 5.15 Ab | 5.53 a | 0.086 | <0.001 | <0.001 | 0.996 |
| | MO | 4.64 Bd | 4.77 ABc | 5.02 ABb | 5.41 a | 0.086 | | | |
| | HO | 4.55 Cd | 4.73 Bc | 4.98 Bb | 5.35 a | 0.089 | | | |
| NH$_3$-N | LO | 3.48 bc | 3.06 Bc | 4.11 Bb | 5.60 a | 0.292 | <0.001 | <0.001 | 0.004 |
| | MO | 3.58 bc | 2.84 Bc | 4.35 Bb | 5.48 a | 0.300 | | | |
| | HO | 3.93 c | 4.89 Ab | 5.42 Aab | 5.89 a | 0.221 | | | |
| LA | LO | 9.82 Aa | 8.30 b | 8.77 Ab | 7.36 Ab | 0.311 | <0.001 | <0.001 | 0.141 |
| | MO | 9.17 Ba | 8.54 b | 6.98 Bc | 6.29 ABd | 0.343 | | | |
| | HO | 9.06 Ba | 7.95 b | 7.09 Bbc | 6.19 Bc | 0.329 | | | |
| AA | LO | 1.02 a | 0.92 Aa | 0.79 Aab | 0.67 Ab | 0.048 | <0.001 | <0.001 | 0.99 |
| | MO | 0.83 a | 0.78 Bb | 0.67 ABb | 0.51 ABc | 0.039 | | | |
| | HO | 0.74 a | 0.67 Cab | 0.59 Bb | 0.43 Bc | 0.037 | | | |
| PA | LO | 1.21 Aa | 0.96 Aab | 0.85 Ab | 0.67 Ab | 0.066 | <0.001 | <0.001 | 0.057 |
| | MO | 0.57 Ba | 0.52 Cab | 0.52 Cab | 0.42 Bb | 0.071 | | | |
| | HO | 0.94 Aa | 0.78 Bb | 0.69 Bb | 0.48 Bc | 0.051 | | | |
| BA | LO | 0.17 b | 0.32 Aa | 0.25 Bab | 0.32 a | 0.021 | 0.706 | <0.001 | 0.003 |
| | MO | 0.24 b | 0.19 Bb | 0.25 Bb | 0.42 a | 0.027 | | | |
| | HO | 0.23 b | 0.24 ABb | 0.29 Ab | 0.38 a | 0.019 | | | |
| CP | LO | 13.48 Aa | 12.81 ab | 12.41 b | 11.53 c | 0.225 | 0.022 | <0.001 | 0.579 |
| | MO | 13.37 Aa | 12.95 a | 12.57 ab | 11.90 b | 0.185 | | | |
| | HO | 12.59 Ba | 12.78 a | 12.25 a | 11.25 b | 0.193 | | | |
| ADF | LO | 29.62 B | 32.6 | 30.44 | 30.66 B | 0.601 | 0.005 | 0.089 | 0.209 |
| | MO | 33.97 Aa | 33.09 a | 30.66 b | 33.79 Aa | 0.462 | | | |
| | HO | 32.76 AB | 32.59 | 32.04 | 34.03 A | 0.368 | | | |
| NDF | LO | 45.79 B | 46.88 B | 45.09 B | 46.74 C | 0.653 | <0.001 | 0.338 | 0.429 |
| | MO | 52.08 A | 50.08 A | 53.56 A | 52.46 B | 0.593 | | | |
| | HO | 55.34 A | 52.48 A | 53.2 A | 56.51 A | 0.766 | | | |

DM, dry matter; CP, crude protein; NDF, neutral detergent fiber; ADF, acid detergent fiber; LA, lactic acid; AA, acetic acid; NH$_3$-N, ammonia nitrogen; PA, propionic acid; BA, butyric acid; pH, the potential of hydrogen. LO, low oat percentages treatment; MO, middle oat percentages treatment; HO, high oat percentages treatment. Different capital letters indicate significant differences between treatments for the same number of silage days ($p < 0.05$); different lowercase letters indicate significant differences between silage days for the same treatment ($p < 0.05$); no or identical letters indicate non-significant ($p > 0.05$). SEM, standard error of the mean; T, treatments; D, aerobic exposure days; T × D, interaction between treatments and aerobic exposure days.

The CP content was significantly decreased with the extension of the aerobic exposure period and was significantly ($p < 0.05$) lower in the HO group than in the LO and MO groups after 3 d of aerobic exposure. The ADF content was significantly ($p < 0.05$) higher in the MO group compared with the LO group after 3 and 12 d of aerobic exposure. Notably, lower ADF content was observed at the aerobic exposure stage of 9 d than in other aerobic exposure stages in the MO group. A lower component of NDF was observed in the LO treatment compared with other groups during the whole aerobic exposure period.

### 3.3. Microbial Compositions of FTMR after Aerobic Exposure

As is shown in Table 3, the interaction of oat addition and aerobic exposure time had a significant effect on LAB content, and the highest amount of LAB was observed in the MO group compared with LO and HO groups after 3, 6, and 9 d of aerobic exposure. The LAB content of LO and HO treatments was higher in the 12 d of aerobic exposure than that in the 3 d of aerobic exposure. Conversely, the LAB content of MO treatment was lower in the 12 d of aerobic exposure than that in the 3 d of aerobic exposure. The amount of AB was decreased in the HO treatment relative to LO and MO treatments after 6 and 12 d aerobic exposure ($p < 0.05$). Notably, the AB content of HO treatment was higher in the 9 d of aerobic

exposure than that in other aerobic exposure times ($p < 0.05$). The yeast content of the three treatments was significantly ($p < 0.05$) increased in 6 d of aerobic exposure and then kept stable in 9 and 12 d of aerobic exposure. Interestingly, the yeast content in the HO group was significantly ($p < 0.05$) higher in 6 d of aerobic exposure than that in the LO group.

**Table 3.** Microbial Compositions of FTMR after Aerobic Exposure.

| Item | Treatment | Aerobic Exposure Days | | | | SEM | *p*-Value | | |
|---|---|---|---|---|---|---|---|---|---|
| | | **AE3** | **AE6** | **AE9** | **AE12** | | **T** | **D** | **D × T** |
| LAB cfu/g FM | LO | 6.29 Aab | 5.13 Bb | 5.38 Bab | 6.82 a | 0.271 | <0.001 | 0.124 | 0.008 |
| | MO | 7.48 Aa | 7.43 Aa | 6.85 Ab | 6.81 b | 0.112 | | | |
| | HO | 4.73 Bb | 6.24 ABa | 5.79 Bab | 6.53 a | 0.265 | | | |
| AB cfu/g FM | LO | 6.82 | 7.22 A | 6.80 | 6.92 A | 0.077 | 0.006 | 0.056 | 0.054 |
| | MO | 6.46 | 6.94 AB | 7.12 | 7.14 A | 0.143 | | | |
| | HO | 6.27 b | 6.06 Bb | 7.05 a | 6.49 Bb | 0.119 | | | |
| Yeast cfu/g FM | LO | 4.38 b | 5.2 Bb | 7.12 a | 7.06 a | 0.358 | 0.241 | <0.001 | 0.599 |
| | MO | 4.15 b | 6.15 ABa | 7.09 a | 7.13 a | 0.381 | | | |
| | HO | 4.54 b | 6.37 Aa | 7.64 a | 7.01 a | 0.382 | | | |

LAB, lactic acid bacteria; AB, aerobic bacteria; LO, low oat percentages treatment; MO, middle oat percentages treatment; HO, high oat percentages treatment. Different capital letters indicate significant differences between treatments for the same number of silage days ($p < 0.05$); different lowercase letters indicate significant differences between silage days for the same treatment ($p < 0.05$); no or identical letters indicate non-significant ($p > 0.05$). SEM, standard error of the mean; T, treatments; D, aerobic exposure days; T × D, interaction between treatments and aerobic exposure days.

### 3.4. Microbial Community of FTMR after Aerobic Exposure

A total of 6,097,489 valid reads were detected, with an average of 81,300 sequences for each FTMR sample. As is shown in Table 4, there was no difference in the Simpson index observed among the three treatments during the whole aerobic exposure period. However, the higher ASV, Shannon, and Chao1 indexes of the terminal ensiling of 60 days were observed in the LO treatment than that in HO and MO treatments. Interestingly, the ASV, Shannon, and Chao1 indexes were higher in the MO treatment relative to HO and LO treatments after 3, 6, 9, and 12 d of aerobic exposure. Notably, the Goods' coverage is more than 99%, indicating that the good coverage of fungi by sample analysis.

**Table 4.** Diversity Indices of FTMR after Aerobic Exposure.

| Days | Treatment | Item | | | | |
|---|---|---|---|---|---|---|
| | | **ASVs** | **Shannon** | **Simpson** | **Chao1** | **Coverage** |
| TE60 | HO | 98.33 c | 3.48 ab | 0.7667 | 99.1267 d | 0.999 |
| | MO | 139.67 bc | 3.13 b | 0.6367 | 142.8333 bcd | 0.999 |
| | LO | 190.00 abc | 4.03 ab | 0.8167 | 192.8200 abcd | 0.999 |
| AE3 | HO | 176 abc | 3.88 ab | 0.7267 | 178.2500 abcd | 0.999 |
| | MO | 184.00 abc | 4.71 ab | 0.8900 | 188.1367 abcd | 0.999 |
| | LO | 161.67 abc | 3.50 ab | 0.6900 | 165.0900 abcd | 0.999 |
| AE6 | HO | 104.33 c | 3.64 ab | 0.8167 | 106.9367 cd | 0.999 |
| | MO | 195.67 abc | 4.86 a | 0.8800 | 199.6600 abc | 0.999 |
| | LO | 181.00 abc | 3.33 ab | 0.6533 | 182.7167 abcd | 0.999 |
| AE9 | HO | 103.33 c | 3.37 ab | 0.7467 | 104.7000 cd | 0.999 |
| | MO | 206.00 ab | 4.59 ab | 0.8633 | 207.5900 ab | 0.999 |
| | LO | 186.00 abc | 4.45 ab | 0.8600 | 188.2333 abcd | 0.999 |
| AE12 | HO | 123.00 bc | 3.39 ab | 0.7033 | 127.3700 bcd | 0.999 |
| | MO | 244.67 a | 4.83 ab | 0.8733 | 249.3000 a | 0.999 |
| | LO | 147.67 bc | 3.88 ab | 0.7567 | 148.2967 bcd | 0.999 |

LO, low oat percentages treatment; MO, middle oat percentages treatment; HO, high oat percentages treatment. Different lowercase letters indicate significant differences between silage days for the same treatment ($p < 0.05$); no or identical letters indicate non-significant ($p > 0.05$). SEM, standard error of the mean; T, treatments; D, aerobic exposure days; T × D, interaction between treatments and aerobic exposure days.

According to the PCoA analysis, the differences in the microbial community structure among three treatments at different aerobic exposure stages were measured. The result displayed that the HO treatment was separated from LO and MO treatments at different aerobic exposure stages (Figure 2).

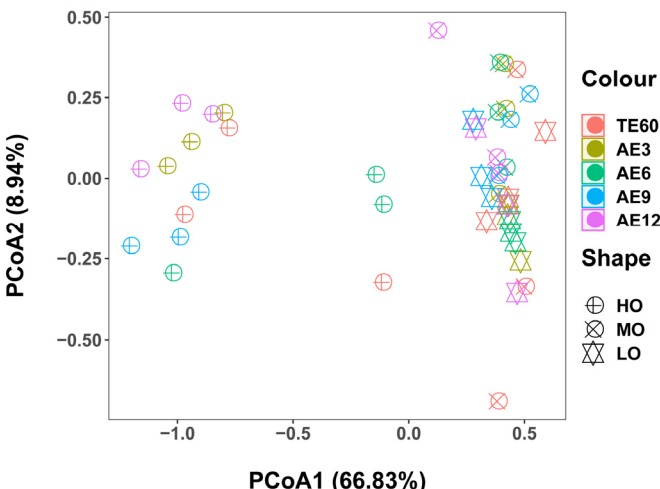

**Figure 2.** Microbial community among different treatments (*n* = 3). Principal coordinates analysis (PCoA) of FTMR samples was performed according to the weighted UniFrac distance. LO, low oat percentages group; MO, middle oat percentages group; HO, high oat percentages group.

At the phylum level, there were four phyla regarded to be the dominant phyla due to the relative abundances of which were more than 1% at least in one group, and the Ascomycota, Basidiomycota, Fundi_unclassified, and Zygomycota were the most abundant phylum (Figure 3A). Notably, the differences in the dominant phyla were absent among the three treatments. At the genus level, the *Eurotiomycetes_unclassified*, *Saccharomyces*, *Wallemia*, *Alternaria*, and *Monascus* were contained in the main genera (Figure 3B). A higher relative abundance of *Saccharomyces* was detected in the HO-AE12 group relative to MO-AE12 and LO-AE12 groups (*p* < 0.05), while the relative abundance of *Eurotiomycetes_unclassified* and *Wallemia* were significantly (*p* < 0.05) lower in the HO-AE12 group than that in MO-AE12 and LO-AE12 groups (Figure 3C).

The LEfSe analysis was selected to reflect the variations of the differences in fungal community structures of the three treatments (LDA score > 3.0). After 60 d of ensiling, the HO-TE60 treatment exhibited high abundances of *Eurotiomycetes_unclassified*, *Sphacelotheca*, and *Basidiomycota_unclassified*, while the LO-TE60 treatment exhibited high abundances of *Malassezia* and *Saccharomyces* (Figure 4A). However, after 3 d of aerobic exposure, the *Athelopsis*, *Mortierellales*, *Sporidiobolus*, and *Tilletia* were mainly enriched in the HO-AE3 treatment, and the *Eurotiomycetes_unclassified* was predominantly found in the MO-AE3 treatment. On the contrary, the *Pichia*, *Volutella*, *Melanocarpus*, *Knufia*, and *Torulaspora* were mainly enriched in the LO-AE3 treatment (Figure 4B). Notably, the MO-AE6 treatment exhibited high abundances of *Trechisporales_unclassified*, *Cyphellophora*, *Knufia*, and *Ascomycota_unclassified*, while the LO-AE6 treatment exhibited high abundances of *Kurtzmanomyces*, *Sphacelotheca*, and *Eurotiomycetes_unclassified* (Figure 4C). Interestingly, the *Talaromyces* and *Uromyces* could predominantly found in the MO-AE9 treatment, and the *Wallemia*, *Malassezia*, *Ceratobasidium*, *Pyrenophora*, and *Debaryomyces* were predominantly found in the LO-AE9 treatment (Figure 4D). After 12 d of aerobic exposure, the *Trechisporales_unclassified*, *Acremonium*, and *Nectriopsis* could predominantly found in the MO-AE12 group, while the *Saccharomyces* and *Torulaspora* were mainly found in the HO-AE12 group (Figure 4E).

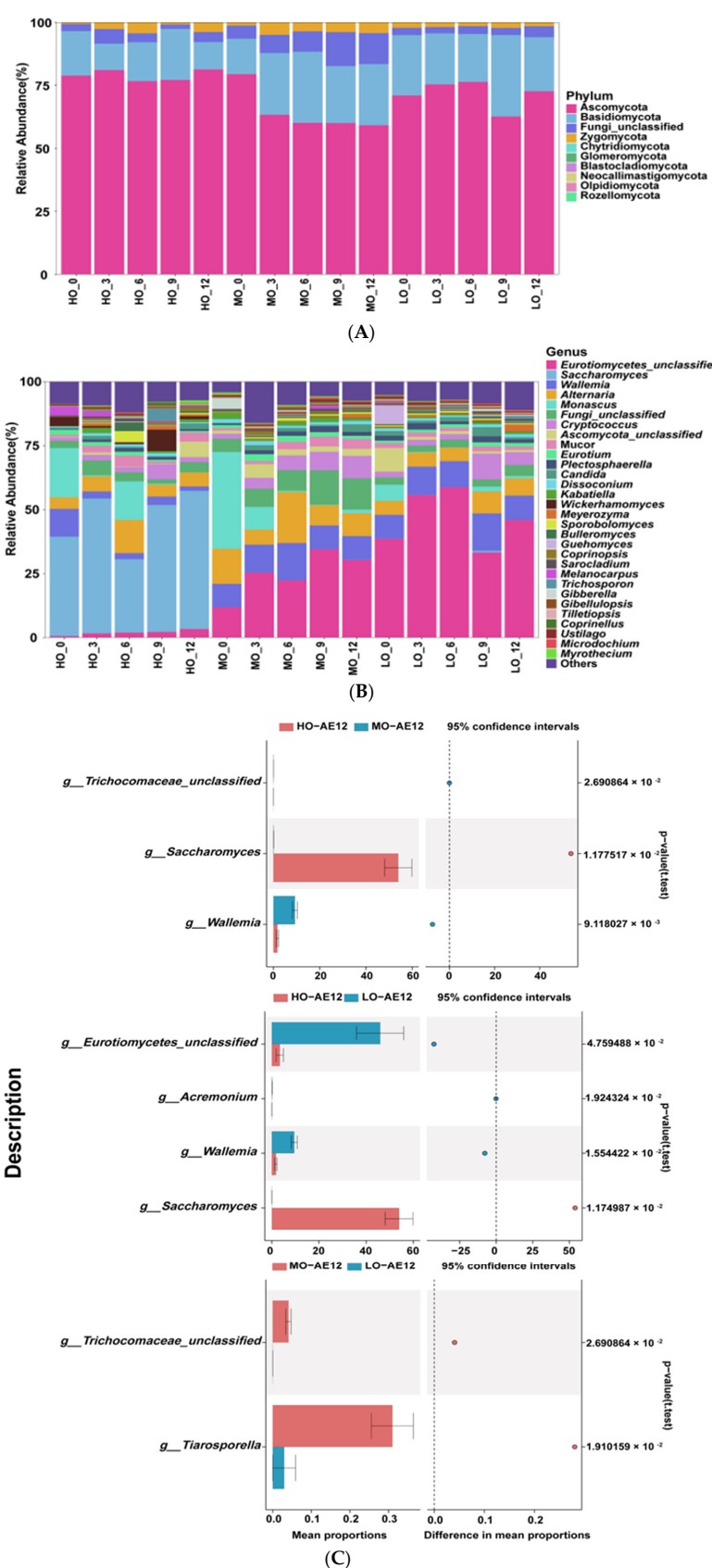

**Figure 3.** The relative abundance (%) of fungal phyla (Top 30) among different treatments (*n* = 3). (**A**) Phylum level. (**B**) Genus level. (**C**) Extended error bar plot showing the fungal at the genus level that had significant differences among the LO, MO, and HO groups. LO, low oat percentages treatment; MO, middle oat percentages treatment; HO, high oat percentages treatment.

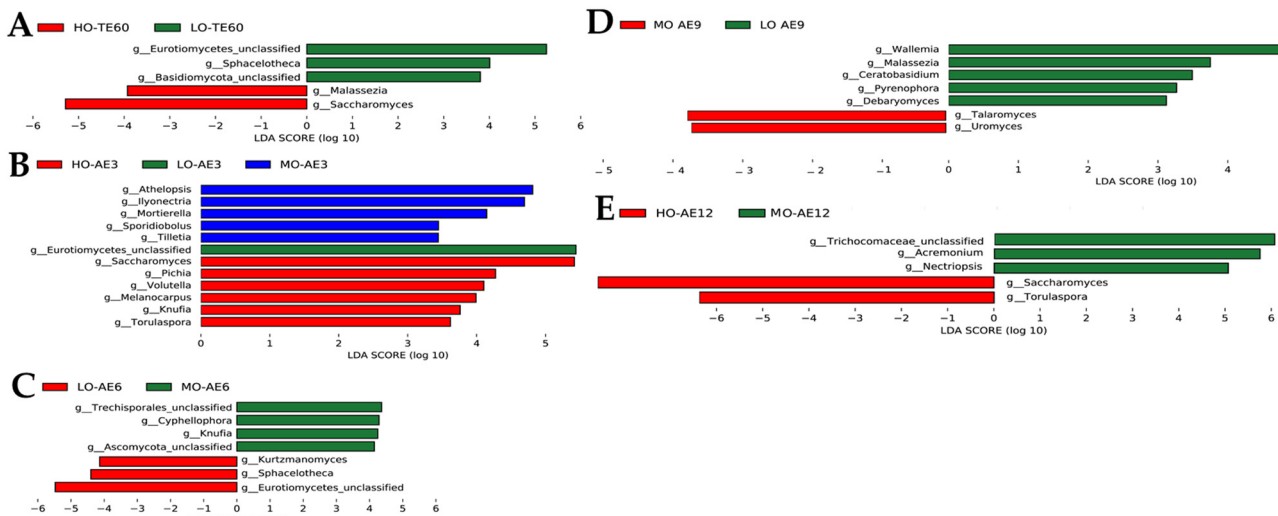

**Figure 4.** Linear discrimination analysis (LDA) of the fungal community of FTMR in the three groups (*n* = 3; default score = 3). (**A**) terminal ensiling of 60 days; (**B**) aerobic exposure for 3 days; (**C**) aerobic exposure for 6 days; (**D**) aerobic exposure for 9 days; (**E**) aerobic exposure for 12 days. The length of the histogram represents the LDA score of different species in the three groups. LO, low oat percentages group; MO, middle oat percentages group; HO, high oat percentages group.

### 3.5. Relationships between Fermentation Characteristics and Fungal Community

In this trial, the studies showed that the LA content was positively related to *Monascus* but inversely associated with *Cryptococcus*. The AA content was positively related to *Wallemia* and *Eurotiomycetes_unclassified* while inversely associated with *Saccharomyces*. The PA content was inversely associated with *Cryptococcus*. The BA content was positively related to *Cryptococcus* while inversely associated with *Melanocarpus* and *Monascus*. The NH₃-N content was positively related to *Saccharomyces* but inversely correlated with *Wallemia*. Finally, the pH value was positively related to *Cryptococcus* and *Eurotiomycetes_unclassified* but inversely associated with *Monascus* (Figure 5).

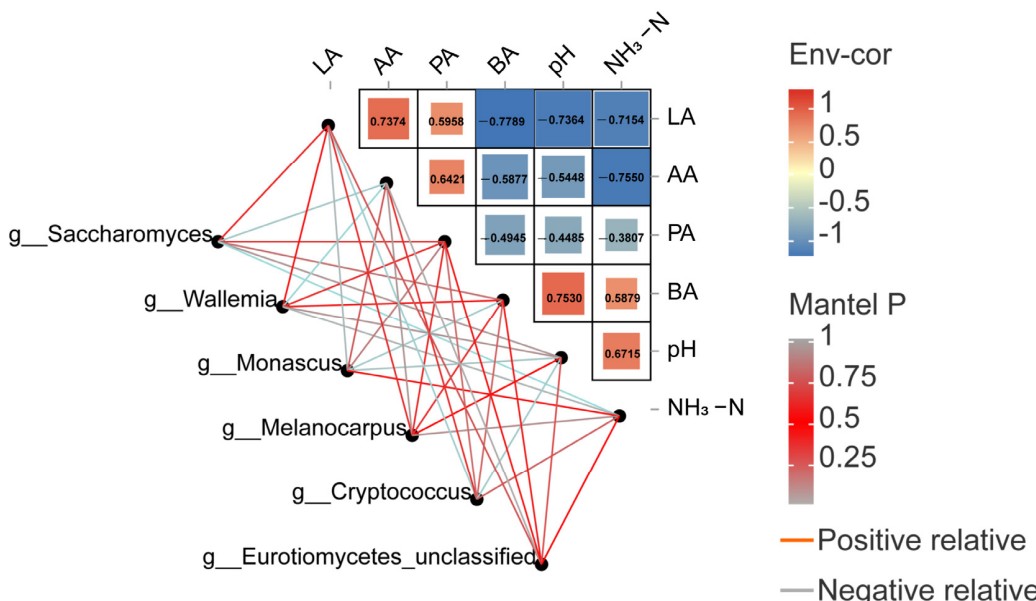

**Figure 5.** Mental test correlation heat map of Pearson's correlations between dominant genera and fermentation parameters. Red represents a positive correlation, while blue represents a negative correlation.

### 3.6. ITS Gene-Predicted Fungal Functional Profiles during Aerobic Exposure Stage of FTMR Analyzed by FUNGuild and PICRUST2

As is shown in Figure 6. A total of 10 main fungal functions were inferred by FUN-Guild. The Saprotroph sections were the primary fungal functional in all FTMR treatments and which continued to the aerobic exposure stage. In addition, the FUNGuild was used to infer the fungal functional groups, suggesting that the plant pathogen and animal pathogen were greatly increased in the HO treatment while greatly decreasing in the MO and LO treatments during the aerobic exposure stage (Figure 6A). The KEGG Module database was used to predict the changes in the key enzymes (top 30) (Figure 6B). The result indicated that the enzymes in the HO group were relatively active during the whole aerobic exposure stage. As is shown in Figure 6C,D, the relative abundances of alcohol dehydrogenase and aldehyde dehydrogenase (NAD(+)) were higher in the HO treatment relative to the other two treatments during the whole aerobic exposure stage.

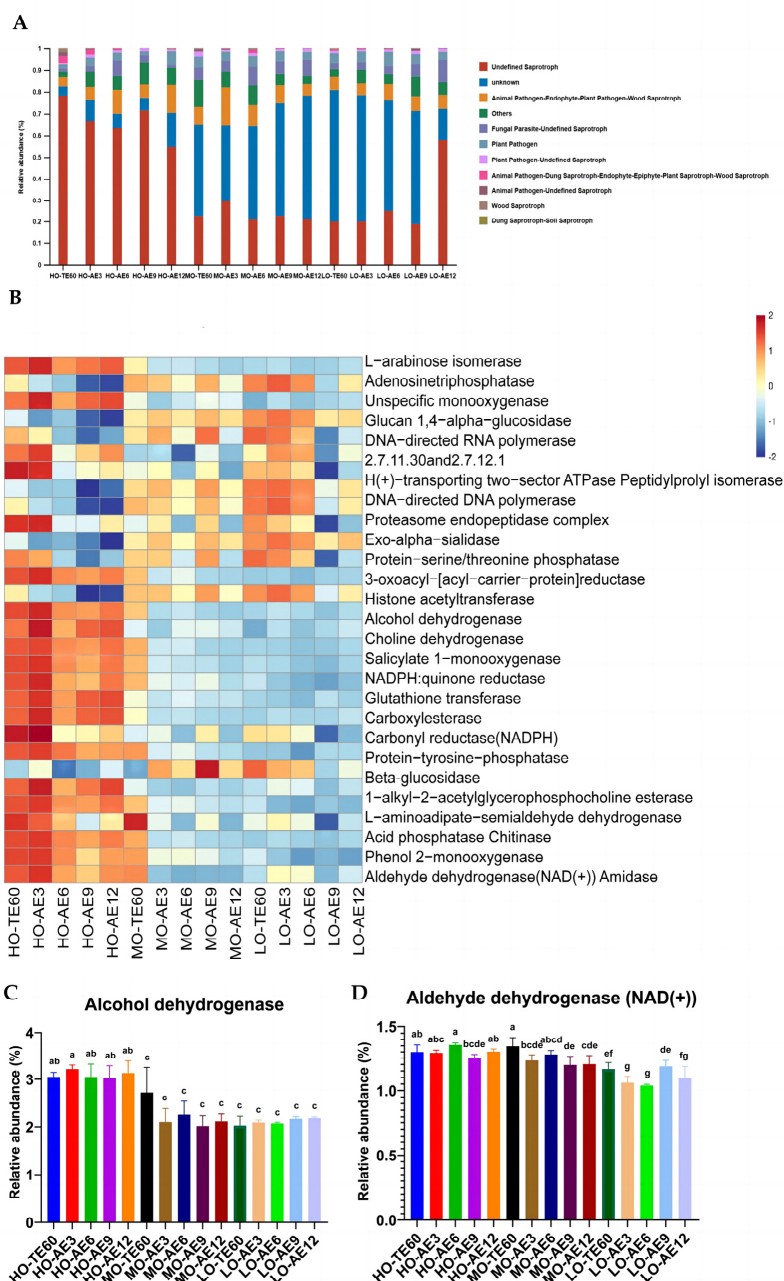

**Figure 6.** ITS gene-predicted fungal functional profiles during the aerobic exposure stage were analyzed by FUNGuild and PICRUSt (*n* = 3). (**A**) Variations in the composition of fungal functional

groups inferred by FUNGuild. (**B**) Level 3 KEGG ortholog functional predictions of the relative abundances of the top 30 enzymes. (**C,D**) Changes of key enzymes involved in fungal community metabolism during the aerobic exposure stage of FTMR. Values with different lowercase (a–g) differ significantly ($p < 0.05$) in (**C,D**).

## 4. Discussion

The results of this research revealed that the aerobic stability was lower in the HO treatment than that in other treatments, suggesting that the higher percentage of oats was unfavorable for the storage of FTMR after aerobic exposure, following a previous trial that the oat could reduce the aerobic stability [30], which may be due to the lower AA content in the HO treatment before aerobic exposure was insufficient to limit the proliferation of fungi, especially yeast [31]. Moreover, Pitt et al. [32] and Salawu [33] reported that the higher buffering capacity and secondary plant metabolites (e.g., saponins) in legumes could inhibit the growth of spoilage microorganisms. Therefore, the lower aerobic stability may also be attributed to the lower alfalfa content in the HO treatment.

Generally, when silage was opened, the aerobic bacteria and yeasts began to activate and proliferate violently, which rapidly consumed lactic acid, causing the pH to increase [34]. Similarly, the pH value in this trial was significantly increased in the three treatments during the whole aerobic exposure stage. The LA content was significantly decreased in the three treatments during the whole aerobic exposure stage, following prior research that the sharp decrease in LA in silage was detected during aerobic exposure [35], which may be attributed to the consumption of lactate by spoilage [16]. Additionally, a lower LA content was always observed in the HO treatment than that in the LO treatment during whole aerobic exposure, which may be due to the higher water-soluble carbohydrates (WSC) content in the HO treatment promoting the proliferation of spoilage microorganisms and their consumption of lactic acid during the aerobic exposure. The aerobic stability of silage is closely related to the level of AA and PA content. In this research, the content of AA and PA significantly decreased with time in all treatments, indicating that the occurrence and aggravation of aerobic corruption during the aerobic exposure stage of FTMR and the aerobic stability in the different treatments was also confirmed by this result. Notably, a higher AA and PA content were observed in the LO treatment relative to other treatments during the aerobic exposure stage of FTMR, which may be because of the differences in the WSC substrates of the three treatments caused by the oat content [36]. More specifically, in the deficiency of WSC, the LA has the potential to be metabolized to AA by acetic acid-producing bacteria [37]. The $NH_3$-N and BA levels increased significantly in all three treatments with increasing aerobic exposure time, which could be explained by the catabolism of proteins and soluble carbohydrates by the proliferation of spoilage microorganisms [38]. Additionally, the interaction of oat addition and aerobic exposure time had a significant effect on $NH_3$-N and BA content, especially at 6 and 9 d of aerobic exposure, with the HO treatment significantly increasing the $NH_3$-N and BA content. This result may be explained by the lower pH value and AA content in HO treatment at 9 and 12 d of aerobic exposure, which is insufficient to inhibit the proliferation of spoilage microorganisms [31], allowing them to compete with *Saccharomyces* for nutrients and accelerating the breakdown of ammoniacal nitrogen and the production of butyric acid [16]. With the prolonging of the aerobic exposure time, the CP content decreased in the three treatments. This finding was consistent with Marián MAJLÁT et al. [39], who reported that the CP content could be degraded in whole-crop wheat silage when exposed to air; this may be that the CP was consumed by the metabolic activity of aerobic microorganisms [31]. The acid hydrolysis during ensiling may affect ADF and DNF content [40]. However, the ADF and NDF content was not affected by aerobic exposure, indicating that the ADF and NDF may not be a substrate for spoilage microorganisms to multiply during aerobic exposure, which may be attributed to the increase in pH value, resulting in the decrease in acidification of ADF and NDF. The results we obtained were in agreement with Liu

et al. [16], who revealed that the components of NDF and ADF were not affected by aerobic exposure time.

Interestingly, the number of LAB in this trial did not reduce significantly and even rose in the LO and HO treatments during the exposure period, suggesting that the LAB could still proliferate and activate violently under aerobic conditions, which was consistent with recent research [41], which observed that the LAB, aerobic bacteria, and yeasts in silage remained more active under aerobic and acidic conditions. In contrast, LAB counts in the MO treatment decreased significantly as the aerobic exposure time was prolonged, which may be due to the higher amount of yeast and AB during the exposure period in the MO treatment, resulting in competition for the same food among microbial populations [42]. Yeast was often the initiator in the aerobic deterioration of silage [43]. In the present trial, the number of yeasts significantly increased during the whole aerobic exposure stage. This finding was consistent with Brüning et al. [44], who observed that the longer the aerobic exposure time, the more critical the yeast population allowed development, and the greater the silage quality deterioration, and the result may be because the yeasts were facultatively anaerobic microorganism, which partly survived the ensiling process during the resting stage and rapidly proliferates re-exposed to air [45]. The presence of AB was also a potential source of increased pH and aerobic spoilage of silage [46]. In this study, the AB content was increased slightly in the three treatments during the whole aerobic exposure stage, which may be explained by the proliferation of yeast increasing the temperature and pH and promoting the reproduction of AB [47]. However, this increase in AB in this trial was not significant, which may be that the time after aerobic exposure measured in this experiment was short, and the survival conditions of aerobic bacteria have not yet reached a stable stage.

Overall, the coverage values of all samples in the three treatments were around 0.99, suggesting that most of the fungal communities were detected. The results of the present trial revealed that the proportion of oat could affect the fungal community composition in FTMR during aerobic exposure. With the prolonged aerobic exposure time, the ASVs number and Chao1 indexes were greater in the HO and MO treatments, which was decreased in the LO treatment. The oat could be easily colonized by mycotoxigenic fungi may be the main reason [48]. As the aerobic exposure time increased, the Shannon index was increased in the MO treatment, while it decreased in the HO and LO treatment, suggesting that the MO treatment has a higher community evenness than other groups.

The PCoA was used to evaluate the difference in the fungal structure and species composition among three treatments during the whole aerobic exposure stage. A clear separation between the HO treatment and the other two treatments showed that the higher percentage of oat exerted an apparent effect on fungal microbial communities.

The phylum-level core microbiomes of FTMR before and after aerobic exposure were Ascomycoda and Basidiomycota, accounting for approximately 90% of fungal species [49,50]. Both Ascomycota and Basidiomycota can decompose cellulose and hemicellulose [51]. Of these, the Ascomycota comprised approximately 11,000 species, which have a broad range of life modes including pathogenic, saprobic, and endophytic, and the Basidiomycota contained more than 40,000 species, which mainly participate in the recycling of nutrients [52,53]. Interestingly, the absence of differences in Basidiomycota and Ascomycota in this research were observed, which may reflect the presence of a core microbiome [54].

Members of the genus *Saccharomyces*, *Monascus*, *Wallemia*, and *Alternaria* were found to be the main microorganisms at the genus level during the aerobic exposure stage, and similar conclusions could be commonly reached in previous studies [55–57]. In addition, some Unidentified fungi, such as *Eurotiomycetes_unclassified*, were also found in the aerobic exposure stage. The *Eurotiomycetes_unclassified* was included in the Eurotiomycetes, which was the class of fungi in the phylum Ascomycota within the kingdom fungi [58]. Although the presence of Eurotiomycetes has been observed in some silage during anaerobic fermentation and aerobic exposure [59,60], the information on silage fungal microbiota from amplicon sequencing is still lacking. Therefore, the roles of *Eurotiomycetes_unclassified* in silage are still unknown, and the result of the LO and MO treatments during the aerobic

exposure stage showing a higher abundance of *Eurotiomycetes_unclassified* was also difficult to interpret. The *Saccharomyces* belonging to lactate-assimilating yeasts have been proven to be associated with aerobic spoilage during aerobic exposure [61], which could utilize lactic acid, resulting in an increase in silage temperature and pH [55]. In the present trial, *Saccharomyces* was the most differentially abundant fungal in the HO treatment during the entire aerobic exposure period, suggesting that the yeast was the most important group of microorganisms causing silage deterioration upon aerobic exposure, which in agreement with the study published by Chen et al. [15], who reported that the presence of yeast could induce the increase in pH value, thereby promoting the proliferation of the spoilage microorganisms. This finding may be attributed to the oat could be easily colonized by mycotoxigenic fungi [50]. Meanwhile, it is also the presence of *Saccharomyces* that explains the phenomenon of low pH and poor aerobic stability in the HO treatment during the aerobic exposure period because the *Saccharomyces* have a high acid resistance [62]. Following the aerobic exposure, the relative abundance of *Wallemia* was increased and gradually replaced the *Eurotiomycetes_unclassified* in the LO treatment. Previous studies have reported that *Wallemia* was the saprotroph fungi, which could reduce the total relative abundance of undesirable fungi [63], suggesting that the higher percentage of alfalfa might inhibit the mold and pathogenic fungi in FTMR during the aerobic exposure period [64], and which might be also explained the higher aerobic stability in the LO treatment. As the aerobic exposure time increased, the coexistence and competition of multiple fungi were observed in the MO group compared with LO and HO groups, suggesting that the interactions of species could result in a specialized and synergic effect that inhibits the reproduction of yeasts and mitigate the role of yeasts in triggering aerobic spoilage [65].

In this study, the *Saccharomyces* were significantly enriched in the HO treatment during the entire aerobic exposure period, which was oppositely correlated with AA content, suggesting that the aerobic spoilage may be caused by *Saccharomyces* [43]. In this trial, the *Wallemia* was positively correlated with AA content, which may inhibit the reproduction of yeast, and the result also explained the phenomenon of higher relative abundance of *Wallemia* and aerobic stability in the LO treatment. Similarly, Hou et al. [58] showed that *Wallemia* could promote acetic acid fermentation. Notably, the *Cryptococcus* was positively correlated with pH value and BA content but inversely correlated with LA and PA content. In contrast, the *Monascus* was inversely related to pH value and BA content, but positively correlated with LA content. In this trial, Both *Cryptococcus* and *Monascus* coexist in the MO treatment. This suggests that the interactions of *Cryptococcus* with *Monascus* may have diminished the dominance of yeast and prolonged aerobic exposure time [65].

In the present trial, the FUNGuild and KEGG Module databases were used to infer the fungal functional groups and the changes in the key enzymes. The result suggested that the higher percentage of oat in FTMR promoted the increase in plant pathogens and animal pathogens. Meanwhile, the multiple key enzymes and metabolic pathways were active in the HO treatment during the entire aerobic exposure period, which allowed the fungi to grow, proliferate, and respond to the environment [66]. Notably, alcohol dehydrogenase and aldehyde dehydrogenase (NAD (+)) could metabolize ethanol and acetaldehyde to promote the production of acetic acid. The relative abundance of alcohol dehydrogenase and acetaldehyde dehydrogenase (NAD (+)) in HO treatment was significantly higher than that in other treatments; this result might be due to the higher proportion of oats in the HO group, providing more initial glucose to promote the metabolism of alcohol dehydrogenase and acetaldehyde dehydrogenase [67]. However, lower AA content was observed in HO treatment, which may be due to the Crabtree effect of high-abundance *Saccharomyces* in the HO group, which preferentially utilizes glucose. Additionally, when glucose is depleted, the ADH2 in *Saccharomyces* is activated under aerobic conditions, accelerating the consumption of ethanol, thereby reducing the production of acetic acid, resulting in a decrease in alcohol dehydrogenase and aldehyde dehydrogenase [68]. Although yeast played a key role in the aerobic spoilage of FTMR in this trial, there is little research on

unknown fungal microorganisms and their related functional genes, which provides us with new research directions and research ideas.

## 5. Conclusions

This preliminary investigation explored the effects of alfalfa hay to oat hay ratios on chemical composition, fermentation characteristics, and fungal communities during aerobic exposure to FTMR. The results suggested that increasing the proportion of oats in FTMR promoted the proliferation of yeast, which reduced aerobic stability by reducing the AA content in FTMR. Further results indicated that FTMR containing higher alfalfa was dominated by *Wallemia* during aerobic exposure, while when the proportion of oats in FTMR reached 30%, the fungal community in FTMR during aerobic exposure was dominated by *Saccharomyces*. Therefore, the proportion of oats in FTMR less than 30% is more conducive to inhibiting the proliferation of spoilage microorganisms and prolonging the aerobic storage time of FTMR. The absence of a bacterial population was the limitation of this trial. Regardless, the findings of this research can provide some references for understanding the interaction between fungal microorganisms and aerobic spoilage in the aerobic exposure phase of FTMR.

**Author Contributions:** M.L. was responsible for conceptualization, methodology, data curation, writing—original draft preparation, and writing—review and editing. L.S. was responsible for methodology. Z.W. was responsible for investigation and resources. G.G. was responsible for writing—review and editing. Y.J. was responsible for project administration and funding acquisition. S.D. was responsible for writing—original draft preparation and writing—review and editing. All authors have read and agreed to the published version of the manuscript.

**Funding:** This work has been partially funded by the National Dairy Technology Innovation Center Creates Key Projects (2021-National Dairy Centre-1).

**Institutional Review Board Statement:** Not applicable.

**Informed Consent Statement:** Not applicable.

**Data Availability Statement:** Sequencing data for 18S rRNA gene sequence have been deposited in NCBI, and sequence read archive databases are publicly available at BioProject (PRJNA947788).

**Acknowledgments:** The authors thank Chaoyue Feed Co., Ltd. (Balin Left Banner, Chifeng, China) for providing the raw materials of the oat and alfalfa.

**Conflicts of Interest:** The authors declare no conflict of interest.

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
