# Peer review of "Effects of Alfalfa Hay to Oat Hay Ratios on Chemical Composition, Fermentation Characteristics, and Fungal Communities during Aerobic Exposure of Fermented Total Mixed Ration"

_fermentation, doi:10.3390/fermentation9050480_

Round 1

Reviewer 1 Report

It was a well-written manuscript, but there are several concerns needed to be addressed before publication.

1) What is the novelty of this work in the field compared to other published studies?

2) What are the objectives of this work? The authors should highlight them in the abstract, introduction section, and describe the findings to eco the objectives.

3) What is the main product generated from aerobic fermentation? 

4) The abbreviations in the abstract should be revised.

5) The experimental design is poor, it is a single-factor design, providing less contribution to the scientific findings.

Author Response

Please see attached for more details.

Reviewer 2 Report

Manuscript 2332439 : Effects of Alfalfa Hay to Oat Hay Ratios on Chemical Composition, Fermentation Characteristics and Fungal Communities during Aerobic Exposure of Fermented Total Mixed Ration.

This manuscript aimed to evaluate the effect of replacing alfalfa hay with oats hay in FTMR on silage quality and fungal communities during the aerobic exposure stage and reveal the interaction mechanism between fungal microorganisms and the silage quality in FTMR during the aerobic exposure stage. The manuscript presents exciting information that may be relevant to a broad audience. However, it has several problems and drawback that prevents publication in its current form. For example, alfalfa is a legume forage with high protein and low energy content. In contrast, oat hay is a grain forage with high energy and low protein content. So, the aim of including these forages in the TMR  is different. The work contains a substantial description of the behaviour of bacterial populations in the silage, but this is not mentioned in the objective, the conclusion, or the title.

Furthermore, while oat inclusion aims to increase the TMR's energy level, alfalfa is to increase the energy content in the ration. The material and method section needs substantial work because the authors assumed that the potential reader knows their previous work, which is very similar to the current manuscript. Having said that, the authors need to improve the description of the different laboratory techniques used. Also, the statistical treatment of results requires a better description because, in the results section, several statistical analysis techniques are mentioned which were not previously mentioned in the method.

The results section must be reduced, it is too large at the moment, and the authors should focus on describing only those variables where significant differences were observed. Furthermore, significant interactions between D* T were presented in tables but not described in the results or the discussion sections. This is particularly relevant because when a significant interaction is observed, it is necessary to focus on explaining the interactions, possibly using interaction plots. The description of interaction is more relevant than that of individual variables. Likewise, the discussion is too large and large chunks of text are devoted to discussing variables like pH where no significant differences were observed.

Individual comments follow:

Abstract

Line 22: State the meaning of LA? There is a figure just below the abstract; delete it.

Line 41: You mean domestic ruminants? Or domestic herbivores?

Line 63: replace affected with “affecting” 

Introduction

Lines 45 to 48: I do not believe it is a good idea to argue that oat hay can replace alfalfa hay. These two ingredients are at the extremes in terms of nutritional quality. However, your introduction may benefit if you expand the information on how the presence of certain species of fungus improves the ruminal degradation of the FTMR, very much like in the objective.

Materials and methods

The description of the chemical and fermentation analyses must be improved; the authors need to mention how many samples were analysed from each FTMR bag and also need to explain briefly how the determination of organic acids was conducted, likewise, the authors need to explain how the microbial analysis was performed. For example, did they isolate all microbes? Or focus on one group of microbes? Also, express all your results on a dry matter basis.

Line 89: Please clarify if the oat hay contained grains and if so, in what percentage?

Line 93: Please define what is the additive in your work? Is the bacteria compound + oat hay? Also, what are the forage samples? Which are the forages?

Line 109: replace with “500 g samples”

Line 120: Did you correct the DM content of the silage to include the volatile solids in the DM?

You may want to check this reference:

HAIGH PM. A note on the relationship between oven and toluene 374 determined dry matter concentrations in Maize silages. Irish J. Agr. Food. Res. 1995; 375 (34): 193-195.

Line 109: How many samples were placed in the bottles?

Line 127: Define “sterile aqueous solution” is it sterile water?

Line 131: Provide a short description of the determination of organic acids. Also, a description of the assessment of the microbial population is necessary.

Line 136: How were the microorganisms enumerated? Did you cultivate for any type, or was there a medium culture selection?

Line 138: Provide information on Man Rogosa_Sharpe agar, malt extract agar etc. Necessary fabricant, country, city. Ideally, catalogue number. What is the meaning of “LAB”.

Line 183: indicate the levels of I and j in the model. Also, show how the difference between treatments was identified. E.g. a Tukey test. Furthermore, I believe that the statistical treatment and analysis of results deserve a better description because it is in the results section that one realises that there were more statistical analyses conducted than those described by the authors.

Results

Line 193: In the methods section, you used days for fermentation time and aerobic exposure, but in the results section, you used hours. You can use any of the two forms but only one. You also used “aerobic deterioration” and “aerobic exposure” Is the meaning the same? It would help if you clarified this.

Figure 1: the description of these figures is unclear, e.g. what is the meaning of “degree of temperature”? Do you mean the time samples take to reach 2 oC above the ambient temperature? The captions in Fig 1B are wrong. Use LO, MO and HO. Why was Ho terminated at approx. 132 h? Why does aerobic exposure time never reach 12 days or 288 hrs?

Table 2: Capital letters to indicate differences between treatments are wrong. Please double-check. I do not see the point of measuring the NDF and ADF content over different aerobic exposure times? It is really more important to determine the digestibility of these fractions because they are exposed to the effects of microbial degradation over time.

Line 223: Avoid using sustainably, remarkably, and markedly expressions to describe results. These are unnecessary expressions.  

Table 2: There are some significant interactions D*T that are not described in this section.

Line : Define all abbreviates in the text; some are only defined in tables.

Table 3: In which units are microbial composition results in this table? Use the same classification of microbial groups in the description of microbial analyses in the materials and method section. What is the meaning of ASVs?

Line 266: You did not mention in the methods section that you will use the Shannon and Simpson indexes.

Lines 280: There was no mention of PCoA analysis in the method section nor in the analysis of results subsection.

 Discussion

Line 370: Define natural compounds content. Do you mean secondary plant metabolites?  

Line 379: What is the meaning of WSC?

Line 375: This sentence contrasts with the previous sentence, where the authors mentioned increased pH. Likewise, there is not much point in discussing pH changes because no significant differences (P<0.05) were observed.

Line 406: Your results do not support this sentence since sugar content was not measured. In fact, I do not believe “sugar” is the best term to use.

Conclusion

The results do not support the conclusion. Moreover, no judgment is made on bacterial population and their role in fermentation.

Author Response

Please see attached for more details.

Round 2

Reviewer 2 Report

The authors have improved the quality of their manuscript and addressed most of my comments. However, there are still a few typos in the new sentences that new English editing and correction. The title of Figure 1 is not clear yet. Please rewrite. The conclusion still needs hard work because it does not conclude the main findings of the work. Line 125 is wrong: 65 oC oven for 48 h?

The author described the significant interactions in the results section but not discussed the implication of the interaction on their results.
